# DIME: Tackling Density Imbalance for High-Performance and Low-Latency Event-Based Object Detection

## Abstract

Event-based object detection takes advantages of the high temporal resolution and dynamic range of event cameras, offering significant benefits in scenarios involving fast motion and challenging lighting conditions. Typically, event streams are first converted into frame sequences through frame-based representations, followed by spatiotemporal feature fusion, similar to video processing. However, video-based processing methods overlook the sparse and non-uniform nature of event streams, making them inadequate for meeting the effectiveness and low-latency processing demands. To address these challenges, we rebuild the spatiotemporal dependency model of event stream by focusing on three key aspects: First, we design a spatiotemporal linear attention to direct build dependencies at patch-level while maintaining spatial parallelism; Second, we incorporate a frame-level temporal decay and spatial position encoding mechanism into the linear attention, which adaptively adjusts the internal state of the network based on the frame information; Third, we propose a structure-level local and global linear attention architecture, which extract event features based on our linear model at different granularities. Our model achieves SOTA performance on Gen1 and 1Mpx datasets, firstly surpassing 50% mAP on 1Mpx with a compact size, while reducing parameters by 3.2× and runtime by 5.1× compared to similar-performing methods, and outperforming lightweight models by +4.3% mAP.

## 1 Introduction

Event cameras have gained significant attention for their ability to record information of changed pixels with microsecond-level resolution and a high dynamic range ($> 120$dB) (Serrano-Gotarredona & Linares-Barranco, 2013; Gallego et al., 2020). They capture fast-moving objects and perform effectively in extreme lighting (bright or dark scenes), providing a spatiotemporal sparse event stream. In contrast, traditional cameras are limited by frame rate, bandwidth, and dynamic range, making them less reliable in such scenarios (Sun et al., 2019; Sayed & Brostow, 2021). As a result, event-based object detection offers low perceptual latency and robust performance, which is crucial for autonomous driving (Gehrig & Scaramuzza, 2024; Shariff et al., 2024).

Since raw event data are difficult to process, a common approach is to convert the event stream into an event frame sequence, which closely resembles the image sequence in RGB video. Consequently, many video processing techniques (Donahue et al., 2015; Arnab et al., 2021; Bertasius et al., 2021) are often adopted in current event-based methods (Perot et al., 2020; Gehrig & Scaramuzza, 2023; Zubic et al., 2024; Peng et al., 2023; Yang et al., 2025) to handle event frames. These video processing approaches typically employ CNNs or softmax attention to extract local or global features, while LSTM or attention to model temporal dynamics. Spatial and temporal computations are then fused either sequentially (late fusion) or within a layer through factorization. This paradigm decouples space and time, enabling independent module optimization and reduced computation.

However, this space-time divided modeling ignores the unique characteristics of event frame data. First, the captured event stream exhibits high temporal resolution, sparsity, and spatiotemporal non-uniformity (Fig.1(a)). Second, the temporal window used to form frames has a variable starting point: as shown in Fig.1(a), windows starting at $t_1$=0 ms and $t_2$=15 ms contain different events,

producing frames with distinct information. Third, the temporal window length may also vary. For example, using 10 ms instead of 5 ms at $t_1$ accumulates more events (Fig.1(b)). As a result, event frames are sparse and have highly uneven information density. In contrast, RGB video is captured at a fixed frame rate (typically 60 fps) by accumulating light intensity, resulting in images that are dense and uniformly distributed. Due to the uneven information density distribution of event frames, the spatiotemporal separation strategy commonly used in video processing becomes sensitive to the information content of "*anchor patches*" (patches that act as intermediate nodes through which spatial and temporal features are fused), making it hard to precisely capture the dependency and leading to suboptimal performance, affecting its high-effectiveness (Fig.1(b)).

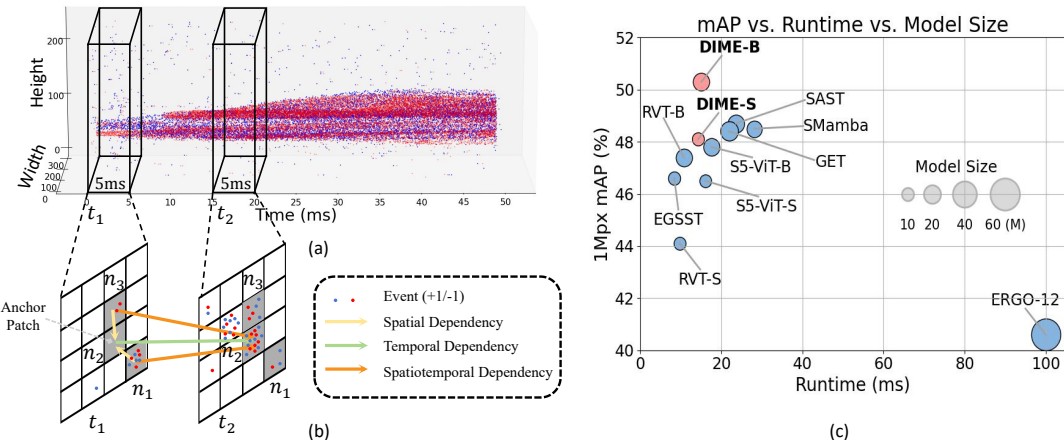

Figure 1: **Characteristics of Event Data and performance on 1Mpx.** (a) The raw event stream exhibits strong spatiotemporal sparsity. (b) Event frames are generated by integrating events within a fixed time window. Each event frame is divided into spatial patches (as defined in Sec.4.1) for subsequent spatiotemporal modeling. $(t_1, n_2)$ is the anchor patch between $(t_1, n_3)$ and $(t_2, n_2)$. (c) On the 1Mpx dataset, our proposed DIME model firstly surpassing 50% mAP with a compact size.

In this work, we propose **DIME** (**D**ensity **I**mbalance **M**itigation for **E**vents) model to address the uneven information density distribution of event frames in object detection. Our approach tackles the problem from three levels. At the patch level, we design a spatiotemporal linear attention model that compresses spatiotemporal information through a hidden state matrix, mitigating the influence of information density at anchor patches and directly constructing dependencies between spatiotemporal patches. Meanwhile, the frame-wise parallel computation scheme ensures real-time causal inference. At the frame level, we design a frame information score based temporal decay and a spatial position encoding module to enable the attention to selectively forget the hidden state matrix and enhance the input features spatially according to the information content of each frame, achieving efficient memory. At the structure-level, we develop a local-global network architecture that progressively extracts event features by building local dependencies in shallow layers and global dependencies in deeper layers.

We validated our model on widely used datasets Gen1 (de Tournemire et al., 2020) and 1Mpx (Perot et al., 2020). And experimental results demonstrate that our model achieves SOTA performance on two datasets and is the first to surpass 50% mAP on the 1Mpx dataset through direct training with a compact model size (Fig.1(c)). Compared to other works with similar performance, our model achieves 3.2× reduction in parameter count and 5.1× reduction in runtime. Moreover, in lightweight network setting, our model significantly outperforms others with comparable parameter size models (+4.3% mAP). Overall, our approach strikes a new balance for the trade-off between performance and efficiency, effectively reducing parameter count and runtime in event-based object detection.

Our contributions can be summarized as follows:

- We highlight the uneven information density of event frames compared to RGB videos, which challenges the effectiveness and low-latency requirements of video-based spatiotemporal modeling in event object detection.

- We introduce **DIME**, which alleviates density imbalance by directly modeling spatiotemporal dependencies, enhancing temporal memory with frame-level information, and balancing feature extraction across local and global scales.

- Our model achieved SOTA performance on the Gen1 and 1Mpx datasets for event object detection, with relatively low runtime and parameter count.

- Even under compact network setting, DIME consistently outperforms alternatives of similar size, underscoring its scalability and practicality for real-time event-based detection.

## 2 RELATED WORKS

**Event-Based Object Detection** leverages the unique advantages of event cameras to excel in high-speed scenarios and challenging lighting conditions. Existing methods are mainly divided into sparse and dense feed-forward approaches. Sparse feed-forward methods, such as Graph Neural Networks (GNNs) (Gehrig & Scaramuzza, 2022; Jeziorek et al., 2023; Sun & Ji, 2023) and Spiking Neural Networks (SNNs) (Luo et al., 2024; Yao et al., 2025; Fan et al., 2024), process sparse event streams with high computational efficiency but often exhibit lower performance and require specialized hardware. For dense feed-forward methods, early approaches converted event stream into image-like tensors for RGB-based detection (Cannici et al., 2019; Lagorce et al., 2014), sacrificing temporal information and hindering slow-motion detection. Later works added recurrent neural network (RNN) layers to capture temporal dynamics and establish spatiotemporal dependencies. For example, RED (Perot et al., 2020) and ASTMNet (Li et al., 2022) use ConvLSTM for spatiotemporal extraction, while RVT (Gehrig & Scaramuzza, 2023) proposes a lightweight backbone that employs spatial softmax attention and temporal LSTM, achieving indirect spatiotemporal modeling. This design achieves SOTA performance with low latency and small model size. Subsequent works largely followed this indirect spatiotemporal modeling approach, enhancing performance and improving computational efficiency by modifying temporal (Zubic et al., 2024) or spatial (Yang et al., 2025; Peng et al., 2024) dependencies or incorporating temporal information in event preprocessing (Peng et al., 2023). However, the uneven distribution of spatiotemporal information density in events may lead to imprecise spatiotemporal dependency modeling, ultimately affecting performance.

**Linear Attention Model.** Linear Transformer (Yang et al., 2023; Qin et al., 2022), SSM (Gu & Dao, 2023; Smith et al., 2022) and Linear RNN (Smith et al., 2022; Qin et al., 2023) address the quadratic complexity of traditional softmax attention, enabling global dependencies with linear computational complexity. Existing works (Chou et al., 2024; Han et al., 2024; Qin et al., 2024) have shown that these models can be unified as a form of linear attention, with three characteristics: **a fixed-size memory space to store contextual information; specific rules to update the memory; equivalent recurrent and parallel forms**. Current researches on linear attention are mostly data-specific. For sequence modeling in NLP, the model needs to handle ultra-long sequence and capture long-range dependencies, thus requiring effective memory mechanism. For example, GLA (Yang et al., 2023) generates input-dependent decay factors through low-rank matrices; Mamba (Gu & Dao, 2023) applies a selective state space model with input-dependent updates and a parallel selective scan for content-aware memory and efficient long-sequence modeling. For static image processing, the focus is on establishing the 2D spatial dependency. Vim (Zhu et al., 2024) flattens the image and applies Mamba for bidirectional scanning to build spatial dependencies; Vmamba (Liu et al., 2024) further extends this to four directions. For video processing, an additional temporal dimension requires spatiotemporal modeling. VideoMamba (Li et al., 2024; Park et al., 2024) embeds the frame sequence into patches, flattens them into a spatiotemporal patch sequence, and applies Mamba with different scanning strategies to capture spatiotemporal dependencies. Our work is built upon the unified form of linear attention models. Considering the uneven distribution of information density in event data and real-time inference demand of event object detection, we design and propose the DIME model, which address the challenges at patch, frame and structure levels.

## 3 CHALLENGES IN EVENT-BASED OBJECT DETECTION

In this section, we describe two challenges in constructing spatiotemporal dependency for event object detection: the uneven distribution of event information density (Sec.3.1) and the real-time requirement in event object detection. (Sec.3.2).

## 3.1 Uneven Distribution Information Density

In event cameras, each event records spatial coordinates, timestamp, and polarity when the light intensity change exceeds a threshold. The original event stream can be represented as:

$$\mathcal{E} = \{e_k | e_k = [x_k, y_k, t_k, p_k]\}, \tag{1}$$

where $x_k, y_k, t_k, p_k$ are spatial coordinates, timestamp and polarity (+1/-1) of the $k$-th event $e_k$. And $\mathcal{E}$ describes motion of objects over time. Due to dynamic triggering and asynchronous output, the raw event stream has uneven information density as Fig.1(a) depicted. Selecting a fixed window of length $\Delta t$ starting at $t$ integrates events into an event frame, as at $t_1$ and $t_2$ in Fig.1(b). The intrinsic information distribution of event stream, together with choices of $t$ and $\Delta t$, will affect the information density distribution in the resulting event frames. Consequently, event frames inherit the sparse and low-level nature of event data, where each position primarily encodes the occurrence of light intensity changes. In contrast, RGB videos are formed by accumulating light over a fixed exposure period, resulting in uniformly dense data where position differences mainly reflect semantic variations (e.g., foreground vs. background). Therefore, the conventional notion of information density in RGB videos cannot be directly applied to event data.

Hence, considering the unique characteristics of event data, we use average event number to measure the information density at each patch in the integrated event frame (Gallego et al., 2020).

$$I(t_i, n_j) = \frac{N_{\mathcal{E}_\mathcal{P}}}{h \times w \times \Delta t}, \tag{2}$$

$$\mathcal{E}_\mathcal{P} = \{e_k | e_k = [x_k, y_k, t_k, p_k], (x_k, y_k) \in \mathcal{P}_{xy}, t_k \in \mathcal{P}_t\}, \tag{3}$$

$$|\mathcal{P}_{xy}| = h \times w, |\mathcal{P}_t| = \Delta t, \tag{4}$$

in which $I(t_i, n_j)$ is the information density at timestamp $t_i$ and position $n_j$, and $N_{\mathcal{E}_\mathcal{P}}$ is the event number of event stream $\mathcal{E}_\mathcal{P}$ within patch $\mathcal{P}$. $h \times w$ is the spatial size and $\Delta t$ is the time window length of the patch $\mathcal{P}$. Therefore, in the Fig.1(b), the information densities $I(t_1, n_1), I(t_2, n_2)$ are relatively high, whereas $I(t_1, n_3)$ is low, and $I(t_1, n_2)$ is even zero since no event occurs. In this case, establishing spatiotemporal dependencies is crucial for extracting target motion from event frames characterized by uneven information density distribution.

Inspired by the processing methods of RGB videos, most of spatiotemporal dependency establishment methods on event frame take an indirect approach (Gehrig & Scaramuzza, 2023; Zubic et al., 2024), where spatial and temporal dependencies are modeled separately (yellow arrows and green arrow in Fig.1(b)) and then aggregated through the anchor patch $(t_1, n_2)$. However, this indirect approach is sensitive to the uneven information density of anchor patch and will lead suboptimal dependencies in event frames. For example, the non-information of anchor patch $(t_1, n_2)$ will result in weak dependencies between $(t_1, n_1)$ and $(t_2, n_2)$ through $(t_1, n_2)$, thereby impairing their overall spatiotemporal dependency. In contrast, the direct spatiotemporal dependency establishment avoids this problem by mitigating the anchor patch, as it in Fig.1(b), directly reflects the continuous relations between $(t_1, n_1)$ and $(t_2, n_2)$ (orange arrow), without being affected by $(t_1, n_2)$.

## 3.2 Causal Real-Time Inference in Event Object Detection

For event-based object detection, causal real-time inference enables each frame to detect targets at the current moment solely based on past information, which is a crucial prerequisite for achieving low-latency object detection (Gehrig & Scaramuzza, 2024; 2023). Some video-based methods directly construct spatiotemporal dependencies, however they are hard to adapt to event object detection due to the real-time requirement. For example, VideoMamba (Li et al., 2024; Park et al., 2024) use a patch-wise recurrent approach with linear attention to build direct spatiotemporal dependency, but introducing causality into non-causal spatial domains, undermining the spatial parallelism and reducing inference efficiency (Han et al., 2024).

## 4 Methods

In this section, we propose **DIME**, a novel event-based framework that integrates three complementary designs across different levels to address the uneven information density in event object detection. Specifically, at the patch-level, we design a spatiotemporal linear attention model that compress

spatiotemporal patch information through a hidden state matrix with real-time inference (Sec.4.1); at the frame level, we introduce a temporal decay based on frame information score (frame-score) and spatial position encoding modules to enhance the linear attention memory (Sec.4.2); at the structure level, we propose a local-global network architecture for efficient event object detection that progressively extracts event frame features (Sec.4.3).

## 4.1 SPATIOTEMPORAL LINEAR ATTENTION

Inspired by the chunk-wise parallelism in GLA (Yang et al., 2023), we design a spatiotemporal linear attention in event object detection. Each event frame at timestamp $t$ is divided into $N$ spatial patches, yielding $\mathbf{X}_{[t]}^f \in \mathbf{R}^{N \times d}$ where $d$ is the feature dimension. Stacking $T$ frames gives $\mathbf{X}^f \in \mathbf{R}^{T \times N \times d}$, which serves as input to our spatiotemporal linear attention. The parallel form is:

$$\mathbf{Q} = \mathbf{X}\boldsymbol{W}_Q, \mathbf{K} = \mathbf{X}\boldsymbol{W}_K, \mathbf{V} = \mathbf{X}\boldsymbol{W}_V \in \mathbf{R}^{(TN) \times d}, \tag{5}$$

$$\mathbf{O} = (\mathbf{Q}\mathbf{K}^T \odot \mathbf{M})\mathbf{V} \in \mathbf{R}^{(TN) \times d}, \tag{6}$$

where $\mathbf{X} \in \mathbf{R}^{(TN) \times d}$ is the input matrix obtained by flattening $\mathbf{X}^f$ across space and time. And $\mathbf{Q}, \mathbf{K}, \mathbf{V}$ are derived through linear mappings with $\boldsymbol{W}_Q, \boldsymbol{W}_K, \boldsymbol{W}_V \in \mathbf{R}^{d \times d}$. $\mathbf{M} \in \mathbf{R}^{(TN) \times (TN)}$ is the causal mask with $\mathbf{M}_{ij} = 1$ if $\lfloor \frac{i}{N} \rfloor \leq \lfloor \frac{j}{N} \rfloor$, else 0. From the Eq.6, $(\mathbf{Q}\mathbf{K}^T \odot \mathbf{M})$ directly establishes the similarity between each patch and all other patches in the spatiotemporal domain (while satisfying causality), yielding a $TN \times TN$ attention score matrix. This matrix is then applied to $\mathbf{V}$ to reweight and recombine information at each spatiotemporal patch and result in the new representation $\mathbf{O}$ that encodes the established dependencies, avoiding the influence of anchor patch in event frame. Its equivalent frame-wise parallel recurrent form is:

$$\mathbf{Q}_{[t]}^f = \mathbf{X}_{[t]}^f \mathbf{W}_Q, \mathbf{K}_{[t]}^f = \mathbf{X}_{[t]}^f \mathbf{W}_K, \mathbf{V}_{[t]}^f = \mathbf{X}_{[t]}^f \mathbf{W}_V, \tag{7}$$

$$\mathbf{S}_{[t]}^f = \mathbf{S}_{[t-1]}^f + \mathbf{K}_{[t]}^{f\,T} \mathbf{V}_{[t]}^f, t = 1, 2, \cdots T, \tag{8}$$

$$\mathbf{O}_{[t]}^f = \mathbf{Q}_{[t]}^f \mathbf{S}_{[t-1]}^f + \mathbf{Q}_{[t]}^f (\mathbf{K}_{[t]}^{f\,T} \mathbf{V}_{[t]}^f) = \mathbf{Q}_{[t]}^f \mathbf{S}_{[t]}^f, \tag{9}$$

where $\mathbf{Q}_{[t]}^f, \mathbf{K}_{[t]}^f, \mathbf{V}_{[t]}^f \in \mathbf{R}^{N \times d}$. $\mathbf{S}_{[t]}^f \in \mathbf{R}^{d \times d}$ is the hidden state, updated by accumulating the history $\mathbf{S}_{[t-1]}^f$ and the incremental information of current input $\mathbf{K}_{[t]}^{f\,T} \mathbf{V}_{[t]}^f$. Therefore, all spatiotemporal dependency information up to $t$ is compressed, memorized, and stored in the state matrix $\mathbf{S}_{[t]}^f$. Multiplying with $\mathbf{Q}_{[t]}^f$ yields the output $\mathbf{O}_{[t]}^f$ that incorporates the historical spatiotemporal dependencies. This frame-parallel computation scheme executes matrix operations for all patches within each frame simultaneously. Moreover, by preserving causal real-time inference across frames, the scheme directly ensures low-latency event object detection.

## 4.2 TEMPORAL DECAY AND SPATIAL ENCODING ENHANCEMENTS

In existing linear attention models, temporal decay is applied token-wise, as inputs are represented as token vectors. However, in our setting, each timestamp corresponds to a frame matrix, making the conventional token-wise temporal decay design incompatible. So, we redesigned the decay by aggregating spatial input into a temporal-scale metric. Given event frames $\mathcal{F} = \{\mathbf{F}_{[t]} \in \mathbf{R}^{H_0 \times W_0 \times C_0}\}_{t=1}^T$, where $T$ is the frame length, $H_0 \times W_0$ is the spatial resolution and $C_0$ is the input channel. We define the frame-score $\mathbf{FS}_{[t]}$ as a weighted sum of variance (motion sharpness) and entropy (structural complexity) (Eq.10), where $\mathbf{FS}_{[t],c}^{\mathrm{var}}$ and $\mathbf{FS}_{[t],c}^{\mathrm{entropy}}$ are computed at

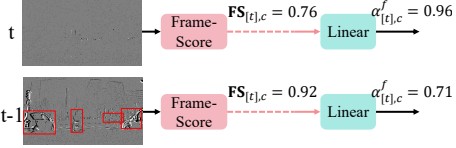

Figure 2: **The frame-score based temporal decay**. "Frame-Score" denotes the frame-score function, while "Linear" is the linear mapping layer. $\alpha_{[t]}^f$ is the temporal decay.

channel $c$, and $\beta$ is a weighting factor. As a result, frames containing richer information are assigned higher scores, which induce stronger decay and greater reliance on the current frame, while frames with less information obtain lower scores, leading to weaker decay and a stronger reliance on historical information (Fig.2). This temporal decay mechanism, which dynamically changes based on the amount of information in the frame, addresses frame-level uneven information density. Comparative

ablation results are presented later and detailed formulas are shown in the appendix.

$$\mathbf{FS}_{[t],c} = \beta * \mathbf{FS}_{[t],c}^{\text{var}} + (1 - \beta)\mathbf{FS}_{[t],c}^{\text{entropy}}. \tag{10}$$

In the spatial domain, we adopt the spatial position encoding method from (Han et al., 2024). In each frame at timestep $t$, encodings like RoPE (Su et al., 2024) serve a similar role to the forget gate in typical linear attention by providing local bias and positional information.The architecture of our linear attention with enhancements is depicted at Fig.3.

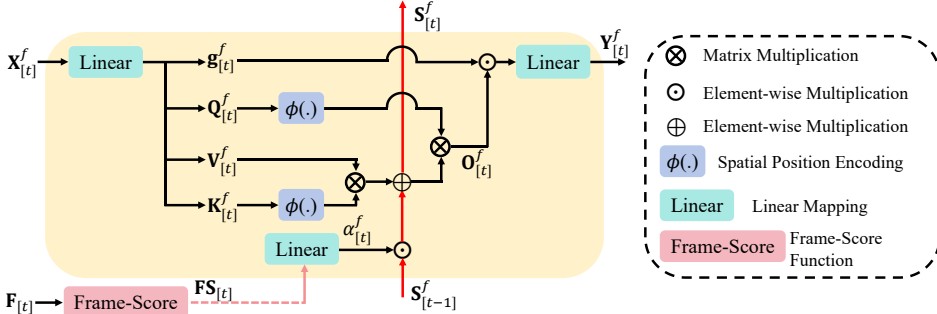

Figure 3: **Recurrent form of spatiotemporal linear attention with enhancements.** It adopts a frame-wise parallel computation scheme, where the input $\mathbf{X}_{[t]}^{f}$ is a frame matrix. Parallel computation is performed within frames through matrix multiplication, while recurrent computation across frames is achieved by updating the hidden state $\mathbf{S}_{[t]}^{f}$. The temporal decay $\alpha_{[t]}^{f}$ is controlled by the frame-score vector $\mathbf{FS}_{[t]}$. And $\mathbf{g}_{[t]}^{f}$ is the output gate.

## 4.3 NETWORK STRUCTURE DESIGN

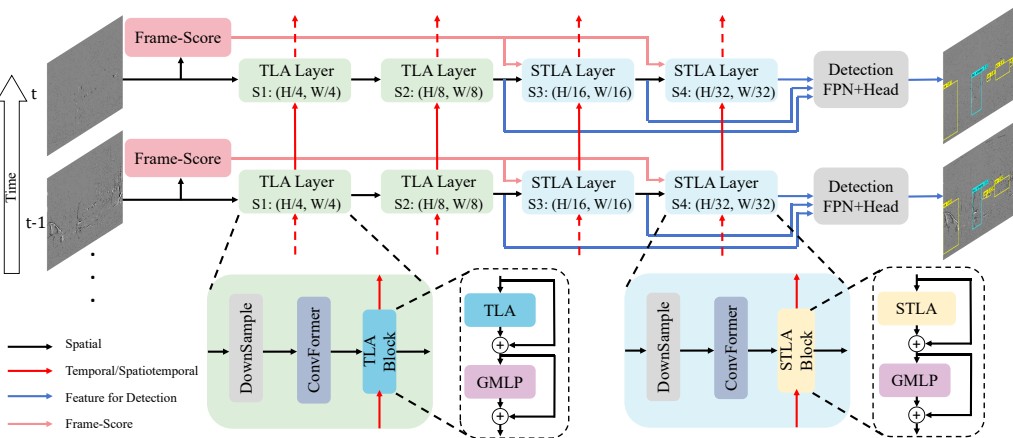

Figure 4: **Overview of our network structure.** The architecture consists of four hierarchical layers. The first two layers employ temporal linear attention (TLA) to capture temporal dependencies, while the latter two layers use STLA to model global spatiotemporal dependencies. Each block follows the MetaFormer design with a token mixer (TLA/STLA) and a channel mixer (GMLP). ConvFormer modules are inserted for local spatial feature extraction, and a convolution-based DownSample module at the beginning of each layer progressively reduces spatial resolution. Frame-Score modules provide temporal decay guidance across layers.

Although events exhibit uneven information density, the spatiotemporal continuity of object motion preserves the local density consistency in early stage. To leverage this property, we adopt a local-global mixing strategy: the first two layers employ temporal linear attention (TLA) (Yang et al., 2023) to capture fine-grained temporal dynamics at the pixel level, while the last two layers adopt spatiotemporal linear attention (STLA) to directly establish global spatiotemporal dependencies.

This deign extracts multiscale spatiotemporal features and mitigates uneven information density at the structure level. To further enhance efficiency, we replace softmax-based global spatial attention with a lightweight convolutional block, ConvFormer (Yu et al., 2023), for local spatial feature extraction. Finally, we follow the MetaFormer design of token-mixer + channel-mixer (Yu et al., 2022), implementing channel-mixer with the Gated MLP (Touvron et al., 2023) (GMLP in Fig.4). Further analysis will be presented in the ablation study. And the network structure of TLA and Gated MLP will be given in the appendix. The complete network structure is shown in Fig.4.

## 5 EXPERIMENT

We validate our methods on the event-based object detection task with Gen1 (de Tournemire et al., 2020) and 1Mpx datasets (Perot et al., 2020), using COCO mAP (mean average precision) (Lin et al., 2014) as the performance evaluation metric. Additionally, we compare our approach with existing works in terms of model size and runtime efficiency.

### 5.1 SET UP

**Datasets.** The Gen1 dataset contains 39 hours of event streams from the Gen1 ATIS $304 \times 240$ sensor (Simon Chane et al., 2016), with 228k car and 27k pedestrian annotations. The 1Mpx dataset provides 15 hours event data at $720 \times 1280$ resolution from the 1-megapixel event camera (Finateu et al., 2020), with 25M bounding boxes for cars, pedestrians, and two-wheelers.

**Experimental Settings.** We follow the experimental settings in RVT. Additionally, we further incorporate the random erasing (Zhong et al., 2020) augmentation method, which randomly select a region from the input frame for erasing. We train our models using two A800 GPUs. Details of our experimental settings and ablation studies about random erasing are provided in the appendix.

### 5.2 BENCHMARK COMPARISONS

We compare our DIME with other works in event-based object detection. We provide two different size of models, DIME-B and DIME-S, with different network's channel settings. Extensive experiments show that our model not only achieves SOTA performance on Gen1 and 1Mpx datasets, but also has a smaller model size and lower runtime. In summary, our approach strikes a new balance between high performance and efficiency. Detailed results are shown in Tab.1.

Table 1: Comparisons on test set of 1Mpx and Gen1 datasets. The best performance is shown in **bold**, and the second-best result is shown with underline. * indicates that the result was reproduced by ourselves under fair experimental conditions. The runtime was tested on RTX 4060ti.

| Methods | Backbone | Params | Gen1 | | 1Mpx | |
|---|---|---|---|---|---|---|
| | | | mAP | Runtime | mAP | Runtime |
| RED (Perot et al., 2020) | CNN+RNN | 24.1M | 40.0 | 16.7ms | 43.0 | 39.3ms |
| ASTMNet (Li et al., 2022) | CNN+RNN | >100M | 46.7 | 35.6ms | 48.3 | 72.3ms |
| ERGO-12 (Zubić et al., 2023) | Transformer | 59.6M | 50.4 | 69.9ms | 40.6 | 100ms |
| RVT-B (Gehrig & Scaramuzza, 2023) | Transformer+RNN | 18.5M | 47.2 | 9.9ms | 47.4 | 10.8ms |
| RVT-S (Gehrig & Scaramuzza, 2023) | Transformer+RNN | 9.9M | 46.5 | 9.5ms | 44.1 | 9.8ms |
| GET-T (Peng et al., 2023) | Transformer+RNN | 21.9M | 47.9 | 16.8ms | 48.4 | 21.9ms |
| SAST-CB (Peng et al., 2024) | Transformer+RNN | 18.9M | 48.2 | 22.7ms | 48.7 | 23.6ms |
| S5-ViT-B (Zubic et al., 2024) | Transformer+SSM | 18.2M | 47.7 | 16.2ms | 47.8 | 17.6ms |
| S5-ViT-S (Zubic et al., 2024) | Transformer+SSM | 9.7M | 46.6 | 14.6ms | 46.5 | 16.1ms |
| EGSST* (Wu et al., 2024) | GNN+LinearViT | 10.4M | 47.5 | **7.8ms** | 46.6 | **8.4ms** |
| SMamba* (Yang et al., 2025) | SSM+RNN | 16.1M | 49.8 | 25.6ms | 48.5 | 28.2ms |
| **DIME-B(ours)** | CNN+LinAttn | 18.6M | **50.8** | 13.6ms | **50.3** | 15.1ms |
| **DIME-S(ours)** | CNN+LinAttn | 10.1M | 50.2 | 12.7ms | 48.1 | 14.3ms |

**Performance.** On the Gen1 dataset, our proposed DIME-B model outperforms the current SOTA with +0.4% mAP improvement, while also achieving better parameter and runtime efficiency. Compared to RED (Perot et al., 2020) and ASTMNet (Li et al., 2022), which also employ CNNs for

spatial feature extraction, DIME-B achieves a substantial gain of +10.8% mAP and +4.1% mAP, respectively. Against models that establish indirect spatiotemporal dependencies like RVT (Gehrig & Scaramuzza, 2023), GET (Peng et al., 2023), SAST (Peng et al., 2024), S5-ViT (Zubic et al., 2024), and SMamba (Yang et al., 2025), our model surpasses them in mAP with a similar model size. Notably, SMamba and SAST strengthen the spatial dependency of RVT, resulting in +2.6% mAP and +1.0% mAP improvement, but this comes with a significant runtime increase (+159% and +129%). In contrast, DIME-B achieves a larger improvement (+3.6% mAP) with only a bit runtime increase (+37.4%), highlighting its efficiency. On the 1Mpx dataset, DIME-B sets a new SOTA with a +1.6% mAP improvement over SAST, becoming the first model to exceed 50% mAP on this dataset. And Fig.5 presents the visualized results of our model on the 1Mpx dataset.

**Model Size.** We further introduce a lightweight variant, DIME-S (10.1M parameters), for fair comparison with compact baselines. Despite its smaller size, DIME-S demonstrates a clear performance gains on both Gen1 and 1Mpx: +4.3% mAP over RVT-S, +4.2% mAP over S5-ViT-S, and +2.7%mAP over EGSST on Gen1; +4.0% mAP, +1.6% mAP, and +1.5% mAP respectively on 1Mpx. Notably, it also surpasses most models with substantially larger parameter counts.

**Runtime.** Although our model has a higher runtime than RVT and EGSST, it shows significant improvements in performance. And compared to other models with better performance, our DIME model demonstrates remarkable advantages in runtime efficiency. For a detailed analysis of the runtime, refer to the ablation experiments of network structure.

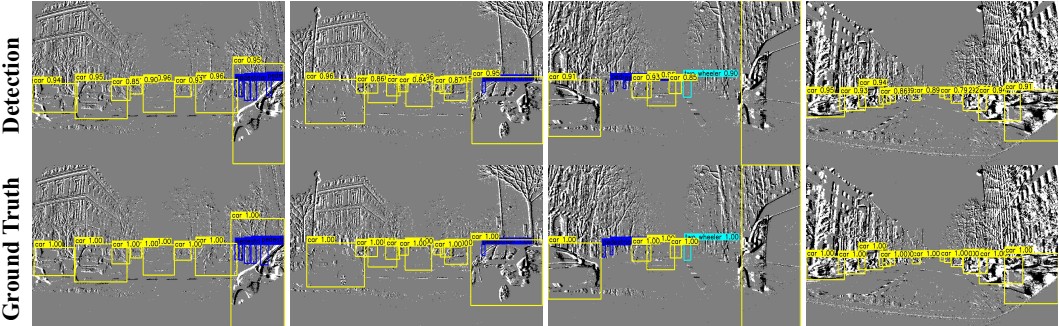

Figure 5: **Visualization of 1Mpx dataset.** We picked some examples with multiple bounding boxes on a single frame, indicating that our model can effectively detect targets even in complex scenes which contain numerous objects. More visualization results can be found in the appendix.

## 5.3 ABLATION STUDY

In this section, we evaluate our contributions from three perspectives. First is the enhancements of temporal decay and spatial encoding. Second is the local-global mixing strategy. Finally, we examine the coupling relationship between temporal decay and spatial feature extraction module. All ablation experiments were conducted on the 1Mpx validation set. Unless specified, spatial feature extraction in the experiments is conducted with MaxViT (Tu et al., 2022) which is a softmax-attention based module used in RVT and S5-ViT.

Table 2: **Spatiotemporal enhancements.**

| Temporal Decay | Spatial Position Encoding | Param(M) | mAP(%) |
|---|---|---|---|
| N/A | N/A | 20.6 | 46.2 |
| Average Pooling | N/A | 20.7 | 47.4 |
| N/A | ✓ | 20.6 | 46.5 |
| Average Pooling | ✓ | 20.7 | **47.7** |
| Frame Score | ✓ | 20.7 | **47.7** |

**Spatiotemporal enhancements.** For comparison, we propose a straightforward design for temporal decay: global average pooling over the spatial dimension of the feature $\mathbf{X}_{[t]}^{f}$, followed by a linear mapping for frame-level decay. As shown in Tab.2, temporal decay gives +1.2% mAP with only 0.1M increase in the model size, and spatial encoding adds +0.3% mAP improvement with negligible

params. Together, they reach 47.7% mAP. When replacing the average pooling with frame-score for temporal decay, it yields no change in model size nor performance. We believe this may be related to the coupling of the spatial feature extraction module and temporal decay, and further ablation experiments will demonstrate this later.

**Local-Global mixing strategy.** Our network follows a local-global mixing strategy: the first two layers use temporal linear attention (TLA) blocks to capture local features, and the last two layers adopt our spatiotemporal linear attention (STLA) blocks to capture global features. To validate this design, we varied the allocation of TLA and STLA across four layers (Tab.3). An STLA-only design yields the lowest performance (43.2% mAP). Increasing TLA modules

Table 3: **Local-Global mixing strategy.** 'T' for TLA, and 'ST' for STLA.

| Layer1 | Layer2 | Layer3 | Layer4 | mAP(%) |
|--------|--------|--------|--------|--------|
| ST | ST | ST | ST | 43.2 |
| T | ST | ST | ST | 45.6 |
| T | T | ST | ST | **47.7** |
| T | T | T | ST | 46.1 |
| T | T | T | T | 46.6 |
| ST | ST | T | T | 43.1 |

from the first layer improved results, with the 'T–T–ST–ST' configuration (ours) achieving the best performance (47.7% mAP). Extending TLA to the third layer caused a drop, and while the TLA-only variant improved over STLA-only, it remained inferior to our design (-1.1% mAP). Swapping the order of TLA and STLA further reduced performance (-4.6% mAP), confirming that local features should be modeled in shallow layers by TLA and global features in deeper layers by STLA.

**Coupling between temporal decay and spatial feature extraction.** In the Tab.2, the addition of the frame-score based temporal decay brings no performance gain. Therefore, we examine it jointly with the spatial feature extraction module, comparing their individual and coupled effects in terms of performance and runtime. Ultimately, we find that the choice of spatial feature extraction directly affects the effectiveness of temporal decay. Results is shown at Tab.4.

Table 4: **The coupling between temporal decay and spatial feature extraction.** ConvFormer has a significant impact on the effectiveness of frame-score based temporal decay.

| Temporal Decay | Spatial Feature Extraction | Params | mAP(%) | Runtime |
|----------------|---------------------------|--------|--------|---------|
| Average Pooling | MaxViT | 20.7M | 47.7 | 12.2ms |
| Average Pooling | ConvFormer | 18.6M | 48.5 | **11.2ms** |
| Frame Score | MaxViT | 20.7M | 47.7 | 16.0ms |
| Frame Score | ConvFormer | 18.6M | **48.9** | 15.1ms |

First, under the average pooling approach, replacing MaxViT with ConvFormer not only reduces the number of model parameters (-2.1M) but also improves model performance (+0.8% mAP) and decreases runtime (-1ms). This likely occurs because DIME's direct spatiotemporal dependency modeling removes the need for heavy spatial softmax attention. Instead, a lightweight convolutional module can be used to extract features from locally consistent information density, further enhancing DIME's ability and ultimately improving detection performance. Then, based on MaxViT, replacing the average pooling method with the frame-score approach does not improve detection performance. However, when ConvFormer is used as the spatial feature extraction module, the frame-score approach can further enhance the model's performance (+0.4% mAP), although it also increases runtime. We believe that the global dependency constructed by MaxViT disrupts the local information density consistency within the event frame, thereby affecting the performance of the subsequent STLA block, while also introducing additional parameter and runtime overhead.

## 6 CONCLUSION

We highlight that, unlike RGB videos, event frames suffer from uneven information density, making video-based spatiotemporal modeling inadequate for effective and low-latency object detection. To address these issues, we propose **DIME**, which aims to mitigate the uneven information density of event frames from three perspectives. At the patch level, we introduce an efficient spatiotemporal linear attention mechanism that directly establishes dependencies while suppressing anchor bias with causal real-time inference. At the frame level, we enhance the model's memory capacity through a frame-score based temporal decay and spatial encoding. Finally, we design a local–global mixing strategy within the network architecture to extract features at multiple granularities, thereby boosting detection performance. Extensive experimental results on event-based object detection datasets demonstrate the effectiveness of our proposed model.

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

# A APPENDIX

## A.1 USE OF LARGE LANGUAGE MODELS (LLMS)

Large Language Models (LLMs) were only used as general-purpose writing assistants in this work. Specifically, we used an LLM to help with language polishing, grammar correction, and improving the clarity and readability of the manuscript. The research problem formulation, method design, experiments, analysis, and all technical contributions were entirely conceived and implemented by the authors. The LLM did not contribute to research ideation, algorithm design, or experimental decision-making, and its role does not rise to the level of authorship or scientific contribution.

## A.2 LIMITATION

There are also some limitations in our work. First, although our method achieves SOTA performance with relatively low latency, there is still a considerable gap compared with works such as RVT, making it difficult to fully meet real-time application requirements. Second, the proposed lightweight model only reduces parameter count and computational cost to some extent, but shows no significant latency improvement. Finally, this work focuses solely on event-based object detection and does not consider modeling strategies that fuse event and RGB data, which can cause the model to fail in low-speed or static scenes. Therefore, optimizing latency and exploring multi-modal modeling with event–RGB fusion will be key directions in our future research. And our codes and models of DIME will be released on GitHub after the review process to contribute to the advancement of the event-based vision community.

## A.3 DETAILS OF MODULES

### A.3.1 TEMPORAL DECAY DESIGN

For the frame-score based temporal decay design, we propose two schemes: one is based on feature average pooling, and the other is based on the weighted variance and entropy of input frames for the design of temporal decay. The input feature at timestamp $t$ is $\mathbf{X}_{[t]}^f \in \mathbf{R}^{N \times d}$, where $N$ is the split patch length (embedded spatial size), $d$ is the mapped channels. And the original input frame of the network is $\mathbf{F}_{[t]} \in \mathbf{R}^{H_0 \times W_0 \times C_0}$, in which $H_0 \times W_0$ is the spatial resolution of event camera and $C_0$ is the input channels. For the feature average pooling-based temporal decay, the formula is:

$$\bar{\mathbf{x}}_{[t]}^f = \frac{1}{N} \sum_{n=1}^N \mathbf{X}_{[t]}^f(n), \tag{11}$$

$$\alpha_{[t]}^f = \bar{\mathbf{x}}_{[t]}^f \mathbf{W}_\alpha, \tag{12}$$

where $\bar{\mathbf{x}}_{[t]}^f \in \mathbf{R}^{1 \times d}$ is feature vector averaged over the spatial dimension, and the temporal decay vector $\alpha_{[t]}^f \in \mathbf{R}^{1 \times d}$ is obtained by the linear transformation matrix $\mathbf{W}_\alpha$.

And the weighted variance and entropy based temporal decay is:

$$\mathbf{p}_{[t],c}(i) = \frac{1}{N} \sum_{n=1}^N \delta(\mathbf{F}_{[t],c}(n) = i), \tag{13}$$

$$\mathbf{FS}_{[t],c}^{\text{entropy}} = (1 - 2 * (-\sum_{i=1}^{E_N} \mathbf{p}_{[t],c}(i) * \log_2 \mathbf{p}_{[t],c}(i))), \tag{14}$$

$$\mathbf{FS}_{[t],c}^{\text{var}} = (2 * \exp(-\text{var}(\mathbf{F}_{[t],c}) - 1)), \tag{15}$$

$$\mathbf{FS}_{[t],c} = \beta \mathbf{FS}_{[t],c}^{\text{var}} + (1 - \beta) \mathbf{FS}_{[t],c}^{\text{entropy}}, \tag{16}$$

$$\alpha_{[t]}^f = \mathbf{FS}_{[t]} \mathbf{W}_\alpha, \tag{17}$$

where $\mathbf{p}_{[t],c}(i) \in \mathbb{R}$ is the probability distribution of events with a count of $i$ at channel $c$, and $\delta(\cdot)$ is the an indicator function that equals 1 when the condition is true, and 0 otherwise. The frame-score

$\mathbf{FS}_{[t],c}$ is composed of two parts, the variance-based $\mathbf{FS}_{[t],c}^{\text{var}}$ and entropy-based $\mathbf{FS}_{[t],c}^{\text{entropy}}$. $\mathbf{FS}_{[t],c}^{\text{var}}$ captures the information of rapidly moving objects over time through variance function $\text{var}(\cdot)$. And $\mathbf{FS}_{[t],c}^{\text{entropy}}$ characterizes the overall structural information through entropy. Both of $\mathbf{FS}_{[t],c}^{\text{var}}$ and $\mathbf{FS}_{[t],c}^{\text{var}}$ are normalized to the range of [-1, 1] and weighted by hyperparameter $\beta$(default 0.5) to obtain the information density $\mathbf{FS}_{[t],c}$. Then, the information density vector $\mathbf{FS}_{[t]} \in \mathbf{R}^{1 \times C_0}$ is passed through a linear transformation for temporal decay $\alpha_{[t]}^f$.

### A.3.2 SPATIOTEMPORAL LINEAR ATTENTION ARCHITECTURE

The computation formula of spatiotemporal linear attention with spatiotemporal enhancements is:

$$\mathbf{S}_{[t]}^f = \text{diag}(\alpha_{[t]}^f)\mathbf{S}_{[t-1]}^f + \phi(\mathbf{K}_{[t]}^f)^T \mathbf{V}_{[t]}^f, t = 1, 2, \cdots T, \tag{18}$$

$$\mathbf{O}_{[t]}^f = \phi(\mathbf{Q}_{[t]}^f)\mathbf{S}_{[t]}^f, \tag{19}$$

$$\mathbf{Y}_{[t]}^f = (\mathbf{O}_{[t]}^f \odot \mathbf{g}_{[t]}^f)\mathbf{W}_O, \tag{20}$$

$$\mathbf{Q}_{[t]}^f = \mathbf{X}_{[t]}^f \mathbf{W}_Q, \mathbf{K}_{[t]}^f = \mathbf{X}_{[t]}^f \mathbf{W}_K, \mathbf{V}_{[t]}^f = \mathbf{X}_{[t]}^f \mathbf{W}_V, \tag{21}$$

$$\alpha_{[t]}^f = \mathbf{FS}_{[t]}\mathbf{W}_\alpha, \mathbf{g}_{[t]}^f = \mathbf{X}_{[t]}^f \mathbf{W}_g, \tag{22}$$

$$\phi(\cdot) = \text{RoPE}(\text{ELU}(\cdot) + 1), \tag{23}$$

where $\mathbf{Q}[t]^f, \mathbf{K}[t]^f, \mathbf{V}[t]^f \in \mathbf{R}^{N \times d}$ denote the query, key, and value matrices, obtained by linear transformation on the input feature $\mathbf{X}^f[t]$ at the $t$-th frame. $\mathbf{S}[t]^f \in \mathbb{R}^{d \times d}$ represents the hidden state matrix. The term $\alpha^f[t]$ is the temporal decay factor based frame-score $\mathbf{FS}_{[t]}$, and $\phi(\cdot)$ is the spatial position encoding function. Furthermore, $\mathbf{g}_{[t]}^f$ is the output gate applied on $\mathbf{O}_{[t]}^f$. After a linear transformation, the final output frame is denoted as $\mathbf{Y}_{[t]}^f$.

### A.3.3 TEMPORAL LINEAR ATTENTION ARCHITECTURE

For the temporal linear attention model, we adopt a simple yet effective Gated Linear Attention (GLA)(Yang et al., 2023), and its recurrent computation formula is:

$$\mathbf{s}_t = \text{diag}(\alpha_t)\mathbf{s}_{t-1} + \mathbf{k}_t^T \mathbf{v}_t, t = 1, 2, \cdots T, \tag{24}$$

$$\mathbf{o}_t = \mathbf{q}_t \mathbf{s}_t, \tag{25}$$

$$\mathbf{y}_t = (\mathbf{o}_t \odot \mathbf{g}_t)\mathbf{W}_o, \tag{26}$$

$$\mathbf{q}_t = \mathbf{x}_t \mathbf{W}_q, \mathbf{k}_t = \mathbf{x}_t \mathbf{W}_k, \mathbf{v}_t = \mathbf{x}_t \mathbf{W}_v, \tag{27}$$

$$\alpha_t = \sigma(\mathbf{x}_t \mathbf{W}_\alpha^1 \mathbf{W}_\alpha^2), \mathbf{g}_t = \mathbf{x}_t \mathbf{W}_g, \tag{28}$$

where $\mathbf{q}_t, \mathbf{k}_t, \mathbf{v}_t \in \mathbf{R}^{1 \times d}$ denote the query, key, and value vectors, respectively, obtained from the input vector $\mathbf{x}_t \in \mathbf{R}^{1 \times d}$ at timestamp $t$ through linear mappings $\mathbf{W}_q, \mathbf{W}_k, \mathbf{W}_v \in \mathbf{R}^{d \times d}$. The hidden state matrix $\mathbf{s}_t \in \mathbf{R}^{d \times d}$ is updated by combining the decayed historical state $\text{dig}(\alpha_t)\mathbf{s}_{t-1}$ with the outer product of $\mathbf{k}_t$ and $\mathbf{v}_t$. The decay vector $\alpha_t$ is obtained via a low-rank projection of $\mathbf{x}_t$ with $\mathbf{W}_\alpha^1 \in \mathbf{R}^{d \times r}$ and $\mathbf{W}_\alpha^2 \in \mathbf{R}^{r \times d}$, where $r < d$. The intermediate representation $\mathbf{o}_t$ is computed by multiplying $\mathbf{q}_t$ with the state matrix $\mathbf{s}_t$, and the final output $\mathbf{y}_t$ is obtained by applying a linear mapping $\mathbf{W}_o$ to $\mathbf{o}_t$ followed by an output gate $\mathbf{g}_t$. For event frames carrying spatiotemporal information, the spatial dimension is folded into the batch dimension to enable parallel processing.

### A.3.4 GATED MLP ARCHITECTURE

Gated MLP is a model that replaces the traditional FFN structure with a gating mechanism for channel mixing, and its computation is formally defined as follows:

$$\text{GMLP}(\mathbf{X}) = (\text{Swish}(\mathbf{X}\mathbf{W}_1) \odot \mathbf{X}\mathbf{W}_2)\mathbf{W}_3, \tag{29}$$

where $\text{Swish}(\cdot)$ is the activation, $\odot$ is the element-wise multiplication, $\mathbf{X} \in \mathbf{R}^{T \times d}$ is the input matrix. Linear mappings $\mathbf{W}_1$ and $\mathbf{W}_2 \in \mathbf{R}^{d \times 3d}$ and $\mathbf{W}_3 \in \mathbf{R}^{3d \times d}$

## A.4 Experiment Details

### A.4.1 Details of Experimental Setting

Our models are trained for 400k iterations using 32-bit precision with the Adam optimizer. The learning rate is managed by a OneCycle learning scheduler, which gradually increases to its maximum during a warm-up phase spanning the initial 5% of the iterations, followed by a cosine decay schedule. Following previous works, we adopted a mixed learning strategy that combines Backpropagation Through Time (BPTT) and Truncated BPTT (TBPTT).

For data augmentation, in addition to standard techniques like random horizontal flipping, zooming in, and zooming out, we also apply random erasing (Zhong et al., 2020), which selects a random region of the input for erasure. To further mitigate overfitting, label smoothing and the drop-path strategy are employed.

The YOLOX detection head is integrated after the backbone network. Our models are trained on the Gen1 dataset using two A800 GPUs, with a batch size of 16, a sequence length of 21, and a learning rate of 2.8e-4. For the 1Mpx dataset, the configuration includes a batch size of 12, a sequence length of 10, and a learning rate of 3.5e-4. The training on the Gen1 dataset takes approximately three days, while on the 1Mpx dataset, it requires around four days.

### A.4.2 Random Erasing Augmentation

Random erasing is a data augmentation technique that randomly selects a region of the input image during training to be "erased" or occluded. This process is designed to simulate real-world scenarios such as occlusion, noise, or partial missing information, thereby enhancing the model's robustness and generalization capability. In our framework, we implement two erasing strategies for the input with $T$ frames: fixed mode and temporal mode (see Fig.B.6). The fixed mode randomly

Table B.5: Ablation Study of Random Erasing data augmentation.

| Random Earse mode | mAP(%) |
|---|---|
| none | 46.4 |
| fixed | 47.7 |
| temporal | 46.6 |

selects a fixed region to erase across all frames, while the temporal mode randomly selects different regions at different timestep. Experimental results in Tab.B.5 demonstrate that random erasing significantly contributes to performance improvement. We believe that the original event may inherently contain less information in each frame. Therefore, applying region suppression at each timestep could potentially lead to target loss over the entire time sequence, thereby affecting the detection results.

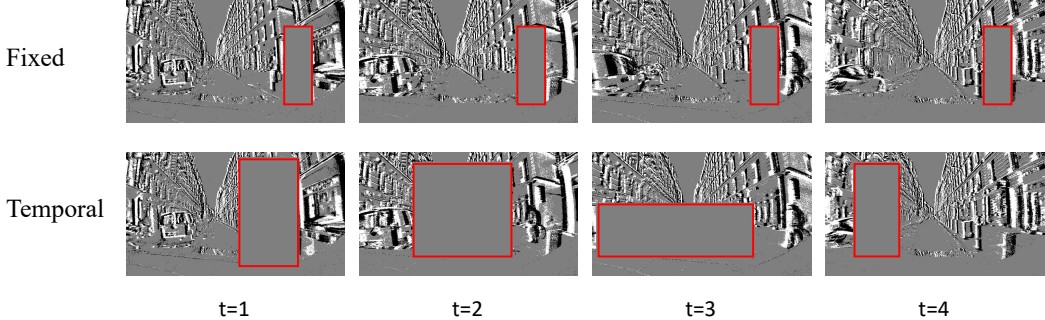

Figure B.6: **Random Erasing.** We provide examples of random erasing for inputs at four different timestamps in both Fixed and Temporal modes, where the red boxes indicate the erased regions. In the first row (Fixed mode), the erasing occurs at the same fixed locations across all timestamps. In the second row (Temporal mode), the erased regions are randomized for each timestamp.

### A.4.3 Detection Visualization

For the visualization results, I have provided more comparisons between the detection results of our model and the ground truth on the 1Mpx and Gen1 datasets, as shown in Fig.B.7. It can be

observed that our model demonstrates excellent detection performance in both information-dense and information-sparse scenarios.

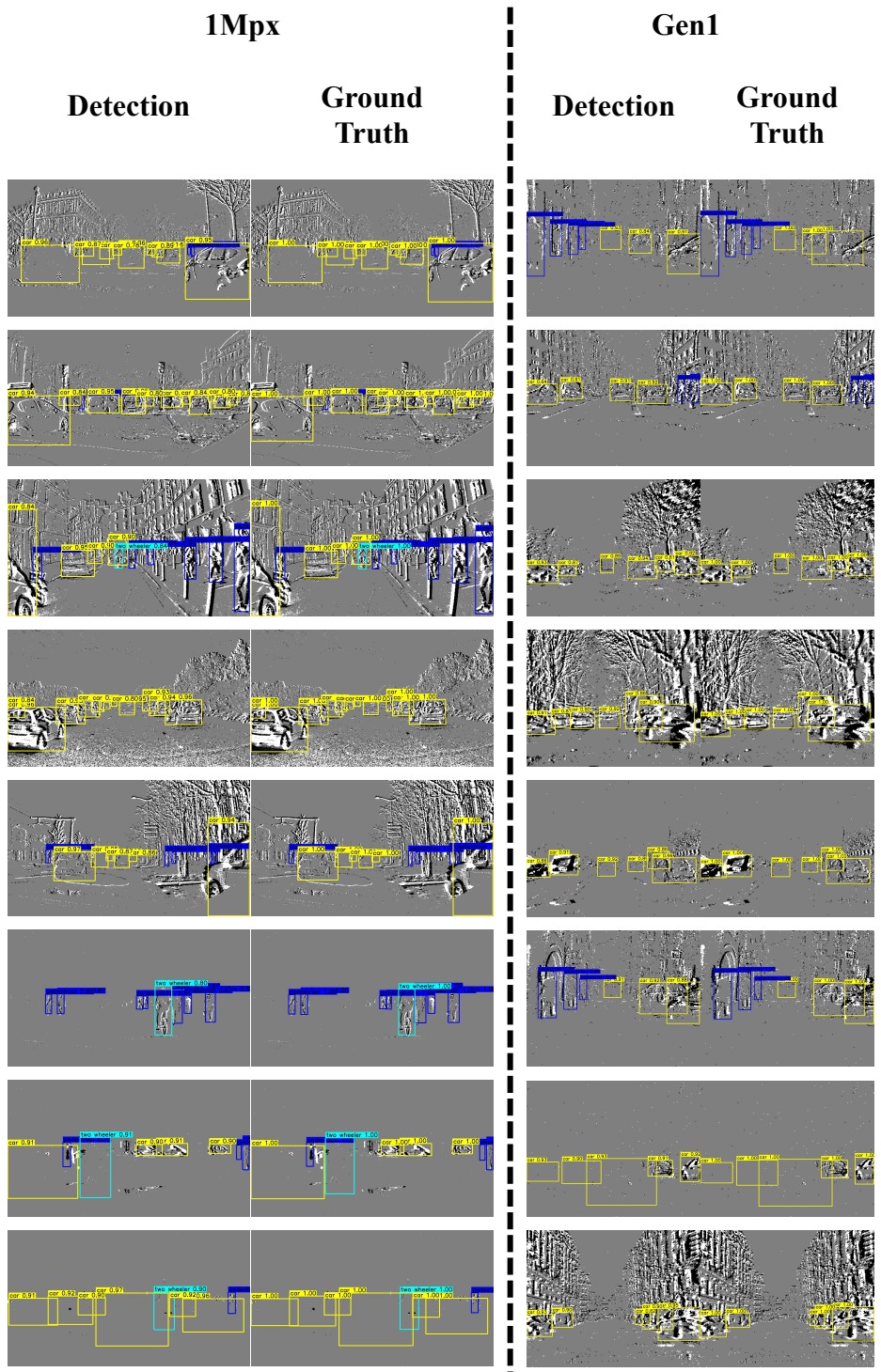

Figure B.7: **Visualization of 1Mpx and Gen1 dataset.** We picked some examples with multiple bounding boxes on a single frame, indicating that our model can effectively detect targets in the scene even when it is complex and contains numerous objects.

