# OpenReview forum: "DIME: Tackling Density Imbalance for High-Performance and Low-Latency Event-Based Object Detection"
_ICLR.cc/2026/Conference — Submitted to ICLR 2026_

### Official Review · Reviewer_h62Z · 2025-10-30

**Soundness:** 2
**Presentation:** 3
**Contribution:** 2
**Rating:** 4
**Confidence:** 4

**Summary:**

This paper presents DIME, a novel method addressing the uneven information density in event frames for event-based object detection. DIME is designed at three levels: (i) a spatiotemporal linear attention mechanism at the patch level to directly model dependencies and avoid anchor-block bias; (ii) a frame-level scoring and temporal decay encoding to enhance adaptive memory of informative regions; and (iii) a local–global hybrid architecture for progressive multi-scale feature extraction. The results show that DIME achieves SOTA performance on the Gen1 and 1Mpx datasets, with notably fewer parameters and faster inference than existing methods.

**Strengths:**

1. The proposed method is thoroughly validated across multiple datasets and compared against a wide range of baselines, demonstrating clear overall advantages in accuracy, parameter efficiency, and inference speed.

2. The lightweight DIME-S significantly reduces model parameters while maintaining high performance, making it well-suited for real-time deployment.

3. The manuscript is clearly written with detailed figures and tables, a complete appendix.

**Weaknesses:**

1. The motivation of this paper is to achieve low-latency and high-performance object detection. From my perspective, this goal could be more effectively addressed through asynchronous fusion of both frames and events, rather than using only a single event modality. Nevertheless, the focus on event-based object detection alone is still a valuable direction. If low latency is the primary objective, approaches such as sparse convolution or graph-based methods might be more appropriate. The idea of addressing density imbalance as the central motivation is not entirely convincing to me. Could the authors provide a clearer explanation of how this directly contributes to latency reduction or performance improvement?

2. In Section 3.1, the authors describe aspects of event density. Are these currently treated as hyperparameters? Could the authors include experiments comparing different fixed temporal window lengths or event counts, and contrast them with the proposed method to better demonstrate its advantage?

Overall, this work represents an incremental improvement in event-based object detection, rather than a major breakthrough. I did not find strong contributions toward areas such as asynchronous event-based detection, event-based multimodal LLMs for detection, or event-based stereo 3D detection. I encourage the authors to carefully consider feedback from other reviewers as well. For now, I rate this paper a 4, and will re-evaluate after the rebuttal and further discussion.

**Questions:**

Please provide responses to each of the comments regarding the weaknesses.

In addition, how could the tracking density imbalance technique be applied solely to event representations, and ideally extended to a joint representation of events and frames? This might make the approach even more useful for the field.

---

### Official Review · Reviewer_PbB3 · 2025-10-30

**Soundness:** 2
**Presentation:** 3
**Contribution:** 2
**Rating:** 4
**Confidence:** 4

**Summary:**

This paper tackles event-based object detection and aims to mitigate event-density imbalance across timestamps and spatial locations with a new method, DIME. DIME consists of a patch-level spatial-temporal attention strategy and a temporal decay method to enhance important patches and frames during training. Experiments on the Gen1 and 1Mps datasets demonstrate the state-of-the-art performance.

**Strengths:**

- The problem that this paper aims to address sounds interesting, the asynchronousness and sparsity nature of event data inherently leads to the information imbalance in different patches and frames, usually presenting a significant challenge in handling blank patches and achieving optimal efficacy.
- This paper is well-written, with the motivation and proposed method well-structured.
- Experiments on well-known benchmarks demonstrate a state-of-the-art performance and a good balance between accuracy and speed.

**Weaknesses:**

- The problem statement is compelling, while the resulting solution seems disconnected from this challenge:
   - It is hard to understand why introducing the linear attention across different patches would mitigate the issue from information imbalance in different event data patches. It is not clearly explained what issue the information imbalance actually introduces. Basically, the proposed spatial-temporal attention is a kind of generic strategy that enhances patches beneficial for the final learning objective, instead of balancing the “imbalance” across patches.
   - On the other hand, the proposed temporal decay is designed as a weighted sum of motion sharpness and the entropy. In the appendix, it can be seen that this metric is composed of the information of rapidly moving objects and the overall structural information. It seems that this strategy may further exacerbate the imbalance across patches and frames, since frames with more objects and higher structural complexities will be enhanced while sparse frames will be further marginalized, right?
- The section on “causal real-time inference in event object detection” is vague. Real-time performance is always a challenge in a broad range of object detection tasks. Given the limited information provided in this section, it is hard to see what is uniquely causal real-time inference here, why video-based methods would struggle to adapt to events, or how DIME’s design specifically addresses that challenge.
- Following the previous point, this paper claims that video-based methods introduce significant computational overhead in event-based object detection, while relevant experimental comparisons and discussions are missing.
- The implemented baselines are lower than the original performance, the performance of EGSST and SMamba in Figure 1 and Table 1 is much lower than the best results in their original paper. It is not clear whether experiments are performed fairly.
- Given the different performance in the ablation results and main results, it seems that different backbones are used. The reasons for the lack of improvement on Maxvit may require further discussion and analysis.
- Reproducibility statement is missing.

**Questions:**

see weaknesses

---

### Official Review · Reviewer_x4Gy · 2025-10-31

**Soundness:** 2
**Presentation:** 2
**Contribution:** 2
**Rating:** 4
**Confidence:** 4

**Summary:**

This paper addresses the problem of spatiotemporal modeling degradation caused by uneven event-frame density in event cameras. The authors propose a novel framework named DIME for event-based object detection, which introduces improvements from three complementary levels. At the patch level, a temporal linear attention mechanism is designed to directly establish cross-spatiotemporal dependencies while avoiding the anchor-patch issue. At the frame level, a frame scoring–driven temporal decay and spatial position encoding scheme is used to enhance the network’s dynamic memory and representation capacity. At the structural level, a local–global hybrid architecture is proposed to enable multi-scale feature extraction. Overall, the framework is conceptually sound and experimentally validated on both the Gen1 and 1Mpx datasets, demonstrating a good balance between accuracy and real-time performance.

**Strengths:**

The introduction of linear attention with spatial positional encoding effectively captures global dependencies while maintaining local consistency.

The proposed local–global hybrid spatiotemporal modeling (TLA + STLA) achieves a strong trade-off between representation capability and computational efficiency.

The experiments are extensive, including results on major datasets and ablation studies covering temporal decay, spatial encoding, and hierarchical modeling.

**Weaknesses:**

The paper claims that linear attention mitigates the problem of uneven event density, but there is no theoretical explanation or visualization to support this claim. A feature distribution or activation map analysis would help verify the mechanism.

Equation (10) introduces a parameter β (set to 0.5 in the supplementary), but the paper does not discuss how β is chosen or how sensitive the model is to this parameter. Adding a sensitivity or ablation study would strengthen the empirical justification.

Although the paper repeatedly states that video-based models such as RVT fail to meet low-latency requirements, the reported inference time for RVT appears shorter than that of the proposed method. This discrepancy should be clarified, and the advantages of DIME beyond latency should be further emphasized.

The innovation boundary is somewhat unclear. The main novelty seems to rely on replacing RNN modules with linear attention. The paper would benefit from a clearer articulation of the conceptual difference and its impact on spatiotemporal modeling.

Finally, the results of EGSST and SMamba in Table 1 (marked with *) are lower than those reported in their original papers. The authors should clarify whether these numbers come from reimplementation under different conditions or if the original results should be used for a fair comparison.

**Questions:**

What is the essential distinction between DIME and video-based models such as RVT? Beyond replacing RNNs with linear attention, does DIME introduce a fundamentally different modeling perspective?

How does linear attention outperform RNNs and state-space models (SSMs) in the context of event stream modeling? Can the authors provide quantitative or qualitative evidence (e.g., sparsity modeling or gradient stability)?

Could the authors provide additional visualization or statistical analysis under different event densities to better demonstrate DIME’s robustness to uneven event distributions?

---

### Official Review · Reviewer_7tHn · 2025-11-03

**Soundness:** 2
**Presentation:** 2
**Contribution:** 2
**Rating:** 4
**Confidence:** 4

**Summary:**

The paper titled “Dime: Tackling Density Imbalance for High-Performance and Low-Latency Event-Based Object Detection” tackled uneven spatiotemporal information flow in event cameras visa three modules: spatiotemporal dependency through token flattening, temporal decay and spatial position encoding for adaptively adjusting frame-memory, and token mixing for local and global feature extraction. It reports event-based object detection results. The work clearly frames the problem statement, shows effective spatiotemporal coupling but limited comparison to state of the art. Issues such as comparison with event-by-event processing methods, corrupted figures, overemphasis on video-baselines, limited evidence/visualization for uneven information distribution, very small gains from the temporal decay modeling, missing information on dataset rates and interpretability need to be addressed in the revision.

**Strengths:**

1.	The paper articulates the problem statement of handling unevenly distributed spatiotemporal data from event cameras and shows ways to approach the issues through experiments using the three proposed modules.
2.	The experiments section shows results comparing to state of the art event camera based detection models.
3.	The proposed coupling of spatio-temporal events through token flattening, while already utilized in literature, is used well in the final architecture.
4.	The authors introduced the notion of using a frame-score to find a temporal decay parameter for fusing time information, and used temporal and spatiotemporal token mixing to relate object level and global features, which is an innovative contribution.

**Weaknesses:**

1.	Some of the figure labels in the paper has become corrupted and needs to be addressed.

2.	The related works and comparison sections need improvement. There are very limited discussions and comparisons with event-by-event processing models that deal with the asynchronous nature of the data, and hence, the naturally considers the uneven event distribution. Examples includes methods based on neuromorphic computing, event-by-event processing with memory-augmented transformers (Kamal, et. al, ICLR'2023, Kamal et. al,. ECCV 2024, Hamaguchi, et. al. CVPR, to name a few), and graph based methods (AEGNN, CVPR'22).  Current literature review focuses more on the limitations of models that perform event aggregation followed by adaptation of image/video based methods.  It is important that authors clearly present their work in the context of asynchronous event-by-event processing based methods. The quantitative comparison of computation, latency, and model size will also be presented with these methods.

3.	The challenges in event-based object detection section discusses the issue about uneven distribution of information density, but this distribution has not been backed by sufficient data to prove the importance of this problem with event camera. The work would benefit from a visualization showing how the uneven nature of the event distribution causes problems in the traditional detection models.

4.	The methods section needs further explanation on the overview of how spatio-temporal tokens are interacting within the proposed architecture through additional visualization.

5.	In the results section, the effect of the temporal decay is not apparent and seems to contribute very little to the overall performance, questioning the need for the added complexity of using this module.

**Questions:**

1. In figure 1-c, what is the runtime for? Is it for event frames with fixed time-lengths, or is it for variable length events? Also, what device is the timing characterization done on? It needs to be disclosed on the performance plot.

2. What are key intuitive differences the proposed approach compared to the existing approaches that natively processes asynchronous event-streams (instead of performing event aggregation)? It is not clear why authors assumed event-aggregation as at the baseline methods and demonstrated advancements over that.

3. How does the proposed approach compare with prior asynchronous and event-by-event processing methods? Please provide comparison off accuracy, parameters, complexity, and latency.

4. What are the data/event rates of the two mentioned datasets? Please add these for better understanding of the nature of the event data.

5.	In Table 1- could the authors specify whether the runtime is per a fixed event frame, or per event? It is important for the readers to understand the actual data rate when comparing runtime. (for example, 10ms inference time in a 30 fps video would make sense, but 10ms inference time per event in a 1000 fps event rate would not be acceptable.)

6.	From the ablation studies in Table-3 and Table-4, the effects of temporal decay and spatio-temporal coupling are not apparent on the uneven information distribution of event cameras. Could authors visualize the intermediate features  study to highlight how the model accumulates uneven information density for different samples (low speed vs high speed)


7. In Figure-4 it is shown that detection decisions are generated on each time step t, t-1… Could the authors describe the advantage of generating detections in each time step compared to taking the decision only at the final time-step after accumulating over all the frames in a sequence?

8.	In section 4.2 – “while frames with less information obtain lower scores, leading to weaker decay and a stronger reliance on historical information” – do the authors mean that a new frame with less information content contributes less to the whole sequence and the historical information contributes more?

---

### Meta-Review · Area_Chair_Un92 · 2026-01-09

**Summary:**

This paper proposes a method that addresses the uneven information density in event frames for event-based object detection. The paper introduces improvements from three complementary levels: patch level, frame level, and structure level.  The problem that this paper aims to address sounds interesting; however, the reviewers raised concerns regarding unclear explanation of the proposed method, insufficient literature review, not-impressive performance, etc., leading unclear novelty of the proposed method. Writing issues were also raised as a concern.  Benefit of replaying RNN modules with linear attention, which seems the main novelty, is not supported even by arguments. The authors, unfortunately, did not submit their rebuttal to resolve the raised concerns.  Therefore, the concerns remain unresolved.  This paper should be rejected, accordingly.

**Reviewer Concerns:**

Poor writing caused concerns on unclear motivation and unclear novelty.  Insufficient literature review in the context of asynchronous event-by-event processing-based methods is also crucial.  Insufficient validation of the proposed method fails to properly support the claims.  Since no rebuttal was provided, all the concerns remain.

**Reviewer Scores:**

All the reviewers would keep their initial scores or give lower scores because of rebuttal absence.

---

### Decision · Program_Chairs · 2026-01-26

Reject